# Vibration control of a class of flexible mechanical systems with output constraints based on partial differential equations

Yuzhi Tang *

School of Mechanical Engineering, Nantong Institute of Technology, Nantong, Jiangsu Province 226000, China

* tangtangyuzhi@yeah.net

**Data availability statement:** All relevant data are within the paper and its Supporting Information files.

## Abstract

Vibration suppression in flexible mechanical systems (FMSs) significantly enhances the precision and stability of equipment, extending its operational life. This technology is extensively applied across various sectors, including aerospace, robotics, and precision manufacturing. This paper introduces a robust control scheme leveraging a Partial Differential Equations (PDEs) model and Barrier Lyapunov Function (BLF), applied through backstepping technology to manage a flexible mechanical system modeled as an Euler-Bernoulli beam with a centrally attached rigid body. The control laws we have formulated are designed to effectively dampen vibrations and rotations, thereby ensuring system stability despite the presence of environmental disturbances. Throughout the control process, the system output consistently stays within the predefined safety limits. Comparative simulations further validate the effectiveness of the proposed control strategy, showing that it can effectively counteract unforeseen disturbances while ensuring that the output remains within the specified constraints.

## 1 Introduction

Flexible mechanical systems (FMSs) consist of mechanical assemblies that incorporate components susceptible to deformation from movement and applied loads [1–3]. Commonly used in applications where material elasticity or a large aspect ratio in structures can induce vibrations, these systems may experience diminished precision, increased wear and tear, and reduced lifespan. Moreover, the noise generated by these vibrations can negatively affect both comfort and operational safety [4,5]. Consequently, investigating methods to suppress vibrations in FMSs is crucial as it not only boosts system stability and operational precision but also prolongs the lifespan of the machinery and enhances overall performance [6,7]. For instance, in robotic arms or aerospace structures, reducing or controlling vibrations ensures higher operational efficiency and better reliability [8–10].

Research on FMSs has often relied on ordinary differential equations (ODEs) to model their dynamics [11–13]. Although ODEs effectively represent systems with single inputs and

**Funding:** This work is supported by the "14th Five-Year Plan" Key Discipline Funding Project of Jiangsu Province (2022-2).

outputs assuming stable system parameters, they struggle to encapsulate critical dynamic features of FMSs due to their inability to incorporate spatial and complex temporal variations. This oversight makes it essential to adopt partial differential equations (PDEs) in modeling FMSs. PDEs offer a richer framework, enabling a detailed representation of the spatial and temporal dynamics crucial for these systems [14,15]. In [16], the Node-former method proposed by Peng et al. integrates Neural Ordinary Differential Equations (NODEs) with the Informer framework. By discarding data outside the acceptable error range and optimizing the encoder-decoder mechanism, it achieves high-accuracy unmanned aerial vehicle (UAV) trajectory prediction even under prolonged data interruption scenarios. Besides, Hang et al. investigated the fault-tolerant control problem of flexible satellites under actuator faults and multiple disturbances. They proposed an adaptive sliding mode fault-tolerant control method based on the Takagi-Sugeno fuzzy disturbance observer (TSFDO) to handle actuator failures, environmental disturbance torques, and elastic modes caused by flexible appendages in attitude control [17]. However, the aforementioned model based on ODEs also has some limitations, including the possibility of control inputs exceeding the physical limits of the system, leading to overflow [18] or saturation [19,20], and numerical instability, which may amplify errors and affect control accuracy.

Output constraints typically define the limits for measurable system outputs such as displacement, velocity, and acceleration, to ensure they remain within predefined thresholds [21–24]. These constraints are influenced by several factors, including the physical limitations of the hardware (such as the maximum range or speed of actuators), safety protocols (which may prevent excessive vibrations or ensure structural stability), and particular operational demands (like the requirement for precise movements within a specific area). Neglecting these constraints in FMSs can lead to decreased reliability and performance, as well as pose safety and economic hazards. Consequently, developing and implementing effective control strategies is crucial for ensuring the stable and secure functioning of these systems. In [25], model predictive control (MPC) was applied to achieve high-speed control and torsional vibration suppression in a drive system with flexible coupling. By using explicit MPC, computational complexity was reduced while maintaining the same performance as traditional MPC, and the simulation results were validated through experiments. In [26], Liu. et al. develop a boundary control strategy based on inversion techniques for flexible unmanned spacecraft systems with input nonlinearity, asymmetric output constraints, and parameter uncertainty. By employing an improved asymmetric barrier Lyapunov function and adaptive neural network control, the strategy ensures effective vibration suppression and precise angle tracking, while maintaining system robustness and stability. In [27], a flexible double-link robotic arm is managed through an adaptive boundary control approach developed using a PDE model. This method proficiently regulates joint positions and dampens elastic vibrations while compensating for uncertainties in parameters. Besides, the asymptotic stability of this system was confirmed through both theoretical analysis and numerical simulation. In the work by [28], a geometric framework is introduced for addressing constrained exterior differential systems on fibered manifolds with n-dimensional bases. This approach reveals how canonical distributions arise naturally within submanifolds of jet bundles. By analyzing systems of first- and second-order PDEs, the method provides insights into the underlying structure of these systems. In contrast to the studies discussed in [26], Cao's research concentrated on the safe and efficient management of multi-six-rotor UAV systems operating in confined environments. This work investigated adaptive trajectory tracking control challenges under asymmetric time-varying output constraints and input saturation conditions. To address these challenges, a dual-loop control strategy was devised, employing a neural network-based adaptive control technique, adaptive sliding mode differentiators, and an event-triggering mechanism. This approach promoted

system stability and enhanced responsiveness, with simulation tests confirming the method's effectiveness [29].

Based on the detailed analysis of FMSs, presented in our study, we have made several significant contributions to the field:

1. We have developed a control scheme that utilizes the PDE model to accurately represent the dynamics of flexible mechanical systems, including those modeled as Euler-Bernoulli beams. This approach is highly effective in complex environments and provides substantial theoretical backing for vibration and rotation suppression, enhancing overall system stability.

2. Our research introduces robust control laws crafted through the use of Barrier Lyapunov Functions (BLF) and backstepping technology. These laws ensure that the system's output remains within predefined safe boundaries, effectively managing system stability even under the challenges posed by actuator faults and external disturbances.

3. Extensive numerical simulations have been performed to validate the efficacy of our proposed control strategies. The results confirm that our control methods are capable of mitigating unwanted displacements and rotations in the system. This ensures the continued stability and reliability of the flexible mechanical system under various disturbance conditions, while adhering strictly to output constraints.

The structure of this paper is organized as follows: Section 2 provides an overview of Hamilton's Principle, introduces the key lemmas utilized throughout this study, and outlines the assumptions made. Section 3 details the objectives of our control strategy, describes the robust controller design utilizing PDEs, and demonstrates its stability through Lyapunov functions. In Section 4, we employ MATLAB simulations to verify the effectiveness of our proposed control approach. The paper concludes with Section 5, which summarizes our findings and suggests avenues for future research.

## 2 Problem statement and preliminaries

### 2.1 Hamilton principle

**Remark 1:** To enhance clarity, the document employs the following notations: $(*)_z$ denotes the first derivative of any function $(*)$ with respect to $z$, represented as $\frac{\partial (*)}{\partial z}$. Similarly, $(*)_{zz}$, $(*)_{zzz}$, and $(*)_{zzzz}$ represent the second, third, and fourth derivatives of $(*)$ with respect to $z$, expressed as $\frac{\partial^2 (*)}{\partial z^2}$, $\frac{\partial^3 (*)}{\partial z^3}$, and $\frac{\partial^4 (*)}{\partial z^4}$, respectively. For time derivatives, $(*)_t$ indicates the first derivative with respect to $t$, $\frac{\partial (*)}{\partial t}$, and $(*)_{tt}$ signifies the second derivative, $\frac{\partial^2 (*)}{\partial t^2}$. These notations facilitate precise mathematical descriptions and computations throughout the manuscript.

**Remark 2:** The robustness of the control strategy in environments with severe disturbances relies on the adaptive capabilities of the BLF, which dynamically adjusts to fluctuating external influences while maintaining system stability. This indicates that our control scheme can effectively maintain system stability and performance even under varying external conditions.

In this section, the HP is employed to formulate the PDE model of the FMSs. It is important to note that the HP is extensively utilized in PDE modeling, and its efficacy has been well-documented in the literature [30,31]. By precisely computing the system's kinetic and potential energies, along with the virtual work, the PDE models for distributed parameter systems are effectively derived.

The HP, as cited in references [32] and [33], is articulated as follows:

$$\int_{t_1}^{t_2} \delta \left( E_k - E_p + W \right) \, \mathrm{d}t = 0 \tag{1}$$

where, $t_1$ and $t_2$ represent two specific points in time, with $t_1 < t < t_2$ defining the operational interval. Here, $\delta$ represents the variational operator. $E_k$ and $E_p$ refer to the kinetic and potential energies of the system, respectively, while $W$ denotes the work performed by the non-conservative forces acting on the system.

The kinetic energy, denoted as $E_k$, for this system is formulated as follows:

$$E_k(t) = \frac{1}{2} m \left[ (\dot{u}_l)^2 + (\dot{v}_l)^2 + (\dot{w}_l)^2 \right] + \frac{1}{2}\rho \int_0^l \left[ (\dot{u})^2 + (\dot{v})^2 + (\dot{w})^2 \right] \mathrm{d}s \tag{2}$$

where the first term corresponds to the kinetic energy of the end-load, while the second term accounts for the remaining kinetic energy. The potential energy of the system is described as follows:

$$\begin{aligned} E_p(t) = &\frac{1}{2} T \int_0^l \left[ (u')^2 + (v')^2 \right] \mathrm{d}s + \frac{1}{2} EI \int_0^l \left[ (u'')^2 + (v'')^2 \right] \mathrm{d}s \\ &+ \frac{1}{2} EA \int_0^l \left[ w' + \frac{1}{2}(u')^2 + \frac{1}{2}(v')^2 \right]^2 \mathrm{d}s \end{aligned} \tag{3}$$

where, the first term represents the elastic potential energy due to tension, the second term accounts for the elastic potential energy resulting from bending, and the third term pertains to additional elastic potential energy.

The virtual work performed by both the system inputs and the disturbances is described as follows:

$$\begin{aligned} W(t) = &\left( u(t) + d_1(t) \right) w\left(l_1, t\right) + \left( \tau(t) + d_2(t) \right) \varphi\left(l_1, t\right) - \gamma_1 \int_0^{l_1} \dot{w}_l(x,t) w_l(x,t) \, \mathrm{d}x \\ &- \gamma_1 \int_{l_1}^l \dot{w}_R(x,t) w_R(x,t) \, \mathrm{d}x - \gamma_2 \int_0^{l_1} \varphi_l(x,t) \dot{\varphi}_l(x,t) \, \mathrm{d}x \\ &- \gamma_2 \int_{l_1}^l \varphi_R(x,t) \dot{\varphi}_R(x,t) \, \mathrm{d}x \end{aligned} \tag{4}$$

where, $d_1(t)$ and $d_2(t)$ represent disturbances affecting the actuators of $u(t)$ and $\tau(t)$, respectively.

By applying the HP as described in [31], [34], the governing equations for FMSs are derived as follows:

$$\rho \ddot{w}_L(x,t) - K w_{Lxx}(x,t) + K \varphi_{Lx}(x,t) + \gamma_1 \dot{\varphi}_L(x,t) = 0 \tag{5}$$

$$I_p \ddot{\varphi}_L(x,t) - EI \varphi_{Lxx}(x,t) - K w_{Lx}(x,t) + K \varphi_L(x,t) + \gamma_2 \dot{\varphi}_L(x,t) = 0 \tag{6}$$

For all $(x,t) \in (0, L_1) \times [0, \infty)$, within the specified spatial and temporal domain,

$$\rho \ddot{w}_R(x,t) - K w_{Rxx}(x,t) + K \varphi_{Rx}(x,t) + \gamma_1 \dot{\varphi}_R(x,t) = 0 \tag{7}$$

$$I_p \ddot{\varphi}_R(x,t) - EI \varphi_{Rxx}(x,t) - K w_{Rx}(x,t) + K \varphi_R(x,t) + \gamma_2 \dot{\varphi}_R(x,t) = 0 \tag{8}$$

For all $(x, t) \in (L_1, L) \times [0, \infty)$, within this spatial and temporal range, and the corresponding boundary conditions are given as follows:

$$M\ddot{w}(L_1, t) + Kw_{Lx}(L_1, t) - Kw_{Rx}(L_1, t) - u(t) - d_1(t) = 0 \tag{9}$$

$$J\ddot{\varphi}(L_1, t) + EI\varphi_{Lx}(L_1, t) - EI\varphi_{Rx}(L_1, t) - \tau(t) - d_2(t) = 0 \tag{10}$$

$$w_L(0, t) = w_R(L, t) = \varphi_L(0, t) = \varphi_R(L, t) = 0 \tag{11}$$

$$w_L(L_1, t) = w_R(L_1, t) = w_0(t) \tag{12}$$

$$\varphi_L(L_1, t) = \varphi_R(L_1, t) = \varphi_0(t) \tag{13}$$

## 2.2 Some useful lemmas and assumptions

**Assumption 1:** It is assume that the external disturbances impacting the system are bounded [35]. Specifically, the upper limits of these disturbances are defined such that $|d_1(t)| \leq \bar{D}_1$ and $|d_2(t)| \leq \bar{D}_2$, where $\bar{D}_1$ and $\bar{D}_2$ are predetermined positive constants.

**Assumption 2:** It is assume that the system's initial conditions are bounded. Specifically, the initial conditions for the trolley are such that $|w_0(0)| < k_1$ and $|\varphi_0(0)| < k_2$, where $k_1$ and $k_2$ represent positive constraints on $w_0(t)$ and $\varphi_0(t)$, respectively, for all $t \in [0, \infty)$. This assumption ensures that the initial states of the system start within specified limits, aiding in the control and stability of the system over time.

**Assumption 3:** The assumption that external disturbances are bounded by a predefined threshold is crucial for maintaining the efficacy of the control strategy under anticipated operating conditions. This foundational assumption delineates a specific range for disturbance handling within the system design, thereby ensuring that the control strategy's effectiveness remains unimpaired by disturbances that exceed expected levels.

**Lemma 1:** For any function $\vartheta(s, t)$ that is continuously differentiable over the interval $[L_1, L_2]$, the following inequalities hold:

$$\int_{l_1}^{l_2} \vartheta(s, t) \, ds \leq 2(l_2 - l_1) \vartheta^2(l_2, t) + 4(l_2 - l_1)^2 \int_{l_1}^{l_2} \vartheta_s(s, t) \, ds \tag{14}$$

$$\int_{l_1}^{l_2} \vartheta(s, t) \, ds \leq 2(l_2 - l_1) \vartheta^2(l_1, t) + 4(l_2 - l_1)^2 \int_{l_1}^{l_2} \vartheta_s(s, t) \, ds \tag{15}$$

**Proof 1:** By applying the method of integration by parts, we obtain the following result:

$$2 \int_{l_1}^{l_2} (s - l_1) \vartheta(s, t) \vartheta_s(s, t) \, ds = (l_2 - l_1) \vartheta^2(l_2, t) - \int_{l_1}^{l_2} \vartheta^2(s, t) \, ds. \tag{16}$$

Then

$$\int_{l_1}^{l_2} \vartheta^2(s, t) \, ds = (l_2 - l_1) \vartheta^2(l_2, t) - 2 \int_{l_1}^{l_2} (s - l_1) \vartheta(s, t) \vartheta_s(s, t) \, ds$$
$$\leq (l_2 - l_1) \vartheta^2(l_2, t) + \frac{1}{2} \int_{l_1}^{l_2} \vartheta^2(s, t) \, ds + 2(l_2 - l_1)^2 \int_{l_1}^{l_2} \vartheta_s^2(s, t) \, ds \tag{17}$$

From inequality (17), we can deduce the following:

$$\int_{l_1}^{l_2} \vartheta^2(s,t)\,\mathrm{d}s \le 2\,(l_2 - l_1)\,\vartheta^2\,(l_2, t) + 4\,(l_2 - l_1)^2 \int_{l_1}^{l_2} \vartheta_s^2(s,t)\,\mathrm{d}s. \tag{18}$$

Therefore, inequality Eq. (15) can be derived using a similar approach.

**Lemma 2:** Let $\psi_1(s,t), \psi_2(s,t) \in R$ with $s \in [0, L]$ and $t \in [0, \infty)$, an inequality holds:

$$\psi_1(s,t)\psi_2(s,t) \le \frac{1}{\gamma}\psi_1^2(s,t) + \gamma\psi_2^2(s,t) \tag{19}$$

where $\gamma > 0$.

**Lemma 3:** Define $v(z,t)$ as a continuous and first order derivative function. Assume that $v(0,t) = 0$ holds for $\forall t \in [0, \infty)$, then we have

$$v^2 \le l \int_0^l v_z^2\,\mathrm{d}z, \quad \forall z \in [0, l] \tag{20}$$

**Lemma 4:** Let $\bar{f}(s,t) \in R$ be a function whose definition domain is $s \in [0, l]$ and $t \in [0, \infty)$, and has the property $\bar{f}(l,t) = 0$ for $\forall t \in [0, \infty)$, then we have

$$\bar{f}^2(s,t) \le l \int_0^l \bar{f}_s^{\,2}(s,t)\,\mathrm{d}s, \quad \forall s \in [0, l]. \tag{21}$$

**Proof 2:** An inner product is defined as

$$\left(\bar{f}_1(s,t), \bar{f}_2(s,t)\right) = \int_0^l \bar{f}_1(s,t)\bar{f}_2(s,t)\,\mathrm{d}s. \tag{22}$$

Using Cauchy-Schwarz inequality [36], we have

$$\left(\int_0^l \bar{f}_1(s,t)\bar{f}_2(s,t)\,\mathrm{d}s\right)^2 \le \int_0^l \bar{f}_1^2(s,t)\,\mathrm{d}s \int_0^l \bar{f}_2^2(s,t)\,\mathrm{d}s. \tag{23}$$

Let $\bar{f}_2(s,t) = \bar{f}_s(s,t)$ and

$$\bar{f}_1(s,t) = H(s-h) = \begin{cases} 0, s < h \\ 1, s \ge h \end{cases}, \quad \text{for } \forall h \in [0, l]. \tag{24}$$

Then it can be obtained that

$$\bar{f}^2(h,t) = \left(\bar{f}(l,t) - \bar{f}(h,t)\right)^2 \le (l - h) \int_0^l \bar{f}_s^{\,2}(s,t)\,\mathrm{d}s \le l \int_0^l \bar{f}_s^{\,2}(s,t)\,\mathrm{d}s. \tag{25}$$

Thus we have $\bar{f}^2(s,t) \le l \int_0^l \bar{f}_s^{\,2}(s,t)\mathrm{d}s, \forall s \in [0, l]$.

**Lemma 5:** ([37,38]) For any positive constants $k_1$ and $k_2$, define the set $Z_1$ as $\{z_1 \in \mathbb{R} : -k_1 < z_1 < k_2\} \subset \mathbb{R}$ and let $\mathcal{N} = \mathbb{R}^l \times Z_1 \subset \mathbb{R}^{l+1}$, both being open sets. Consider the dynamical system described by the differential equation $\dot{\eta} = h(t, \eta)$, where $\eta = \begin{bmatrix} w \\ z_1 \end{bmatrix}$ is a state vector in $\mathcal{N}$, and the function $h : \mathbb{R}^+ \times \mathcal{N} \to \mathbb{R}^{l+1}$ is piecewise continuous with respect to time $t$ and locally Lipschitz in the state variable $z$, uniformly over time, within $\mathbb{R}^+ \times \mathcal{N}$. Assume

the existence of functions $U : \mathbb{R}^l \to \mathbb{R}^+$ and $V_1 : Z_1 \to \mathbb{R}^+$, both continuously differentiable and positive definite within their respective domains, with $V_1(z_1)$ tending to infinity as $z_1$ approaches $-k_1$ or $k_2$. Additionally, let $U(w)$ be bounded by class $K_\infty$ functions $\gamma_1(\|w\|)$ and $\gamma_2(\|w\|)$. Define the Lyapunov candidate $V(\eta) = V_1(z_1) + U(w)$. If $z_1(0)$ is initially within the interval $(-k_1, k_2)$ and the derivative $\dot{V} = \frac{\partial V}{\partial \eta} h \leq 0$ holds, then $z_1$ will remain within the open interval $(-k_1, k_2)$ for all $t \geq 0$. This confinement is ensured by the properties of the function $V$ and the negative semidefinite condition on $\dot{V}$.

# 3 Design of robust controller and stability proof

## 3.1 Control objective

The primary goal of the robust control strategy developed for the FMSs is to effectively mitigate internal vibrations and stabilize rotational motions. The system is specifically modeled as an Euler-Bernoulli beam with a rigid body attached at its center. The control laws, derived from a PDE model and a BLF, are implemented using backstepping techniques. This approach is designed to ensure that the system's output stays within predefined safety limits. Ultimately, the control scheme aims to achieve stable operation that is resistant to disturbances, reducing any displacement caused by unexpected external forces. By doing so, the system ensures that all movements are kept within acceptable thresholds, thereby preserving the equipment's functionality, integrity, and long-term reliability.

## 3.2 Design of robust controller

**Remark 3:** The Barrier Lyapunov Function (BLF) serves as a critical tool for stability analysis and control design, especially useful in systems where outputs must adhere to specific constraints [37,39,40]. BLF, a variant of the traditional Lyapunov function, is tailored to manage output constraints effectively, ensuring that the system maintains stability while conforming to these limits. It is commonly employed in the development of control laws that stabilize the system amidst external disturbances or uncertainties and maintain outputs within set boundaries. This functionality is crucial in high-stakes applications demanding exceptional safety and precision, such as in spacecraft, robots, and RMs.

To begin with, a transformation of the coordinate system is designed:

$$\eta_1(t) = w_0(t) \tag{26}$$

$$\eta_2(t) = \varphi_0(t) \tag{27}$$

$$\eta_3(t) = \dot{\eta}_1(t) + \varpi_1(t) \tag{28}$$

$$\eta_4(t) = \dot{\eta}_2(t) + \varpi_2(t) \tag{29}$$

where $\varpi_1(t)$ and $\varpi_2(t)$ represent the virtual control laws that have been formulated as follows:

$$\varpi_1(t) = \frac{\alpha}{\beta}\eta_1(t) \tag{30}$$

$$\varpi_2(t) = \frac{\alpha}{\beta}\eta_2(t) \tag{31}$$

where the parameters $\alpha$ and $\beta$ are both positive constants.

Then, a BLF is chosen to be

$$V_a(t) = \frac{\beta}{2} k_1 \eta_1^2(t) + \frac{\beta}{2} k_2 \eta_2^2(t) \tag{32}$$

where $k_1$ and $k_2$ are positive constants, the time derivative of this function is then computed as follows:

$$\dot{V}_a(t) = \beta k_1 w_0(t) \dot{w}_0(t) + \beta k_2 \varphi_0(t) \dot{\varphi}_0(t) \tag{33}$$

To maintain output constraints, a BLF [37,39,41] is incorporated into the Lyapunov function, as follows:

$$V_1(t) = V_a(t) + \frac{\beta}{2} M \eta_3^2(t) a(t) + \frac{\beta}{2} J \eta_4^2(t) b(t) \tag{34}$$

where $a(t) = \ln \frac{2k_1^2}{k_1^2 - w_0^2(t)}$ and $b(t) = \ln \frac{2k_2^2}{k_2^2 - \varphi_0^2(t)}$.

Upon deriving $V_1(t)$ with respect to time, the following is obtained:

$$
\begin{aligned}
\dot{V}_1(t) =& \dot{V}_a(t) + \beta M \eta_3(t) \dot{\eta}_3(t) a(t) + \beta J \eta_4(t) \dot{\eta}_4(t) b(t) \\
& + \frac{\beta}{2} M \eta_3^2(t) \dot{a}(t) + \frac{\beta}{2} J \eta_4^2(t) \dot{b}(t) \\
=& \beta \eta_3(t) \left( -K w_{lx}(l_1, t) + K w_{Rx}(l_1, t) + u(t) + d_1(t) + M \frac{\alpha}{\beta} \dot{w}_0(t) \right) a(t) \\
& + \beta \eta_3(t) \left( -EI \varphi_{lx}(l_1, t) + EI \varphi_{Rx}(l_1, t) + \tau(t) + d_2(t) + J \frac{\alpha}{\beta} \dot{\varphi}_0(t) \right) b(t) \\
& + \beta M \eta_3^2(t) \frac{w_0(t) \dot{w}_0(t)}{k_1^2 - w_0^2(t)} + \beta J \eta_4^2(t) \frac{\varphi_0(t) \dot{\varphi}_0(t)}{k_2^2 - \varphi_0^2(t)} + \beta k_1 w_0(t) \dot{w}_0(t) \\
& + \beta k_2 \varphi_0(t) \dot{\varphi}_0(t)
\end{aligned}
\tag{35}
$$

Let

$$
\begin{aligned}
\mu_1(t) =& k_3 \eta_3(t) + \bar{D}_1 \,\mathrm{sgn}\,(\eta_3(t)) - A(t) + M \frac{\alpha}{\beta} \dot{w}_0(t) \\
& + \frac{k_5 \eta_3(t) + k_1 w_0(t) + A(t)}{a(t)} + \frac{M \eta_3(t)}{a(t)} \frac{w_0(t) \dot{w}_0(t)}{k_1^2 - w_0^2(t)}
\end{aligned}
\tag{36}
$$

$$
\begin{aligned}
\mu_2(t) =& k_4 \eta_4(t) + \bar{D}_2 \,\mathrm{sgn}\,(\eta_4(t)) - B(t) + J \frac{\alpha}{\beta} \dot{\varphi}_0(t) \\
& + \frac{k_6 \eta_4(t) + k_2 \varphi_0(t) + B(t)}{b(t)} + \frac{J \eta_4(t)}{b(t)} \frac{\varphi_0(t) \dot{\varphi}_0(t)}{k_2^2 - \varphi_0^2(t)}
\end{aligned}
\tag{37}
$$

Considering that $k_3, k_4, k_5, k_6 > 0$, and defining $A(t) = K w_{Lx}(l_1, t) - K w_{Rx}(l_1, t)$ and $B(t) = EI \varphi_{lx}(l_1, t) - EI \varphi_{Rx}(l_1, t)$, we obtain the following expression:

$$\dot{V}_1(t) \leq \beta \eta_3(t)$$

$$\left( -k_3 \eta_3(t) + \mu_1(t) - \frac{k_5 \eta_3(t) + k_1 w_0(t) + A(t)}{a(t)} - \frac{M \eta_3(t)}{a(t)} \frac{w_0(t) \dot{w}_0(t)}{k_1^2 - w_0^2(t)} + u(t) \right) a(t)$$

$$+ \beta \eta_4(t) \left( -k_4 \eta_4(t) + \mu_2(t) \right.$$

$$\left. - \frac{k_6 \eta_4(t) + k_2 \varphi_0(t) + B(t)}{b(t)} - \frac{J \eta_4(t)}{b(t)} \frac{\varphi_0(t) \dot{\varphi}_0(t)}{k_2^2 - \varphi_0^2(t)} + \tau(t) \right) b(t) \qquad (38)$$

$$+ \beta M \eta_3^2(t) \frac{w_0(t) \dot{w}_0(t)}{k_1^2 - w_0^2(t)} + \beta J \eta_4^2(t) \frac{\varphi_0(t) \dot{\varphi}_0(t)}{k_2^2 - \varphi_0^2(t)} + \beta k_1 w_0(t) \dot{w}_0(t) + \beta k_2 \varphi_0(t) \dot{\varphi}_0(t)$$

$$= -\beta k_3 \eta_3^2(t) a(t) + \beta \eta_3(t) (\mu_1(t) + u(t)) a(t) - \beta k_4 \eta_4^2(t) b(t)$$

$$+ \beta \eta_4(t) (\mu_2(t) + \tau(t)) b(t) + D$$

where

$$D = - \beta \eta_3(t) (k_5 \eta_3(t) + k_1 w_0(t)) - \beta \eta_3(t) A(t) - \beta \eta_4(t) (k_6 \eta_4(t) + k_2 \varphi_0(t))$$

$$- \beta \eta_4(t) B(t) + \beta k_1 w_0(t) \dot{w}_0(t) + \beta k_2 \varphi_0(t) \dot{\varphi}_0(t) \qquad (39)$$

$$= - k_5 \beta \eta_3^2(t) - \alpha k_1 w_0^2(t) - \beta \eta_3(t) A(t) - k_6 \beta \eta_4^2(t) - \alpha k_2 \varphi_0^2(t) - \beta \eta_4(t) B(t)$$

The control inputs are designed as follows:

$$u(t) = -\mu_1(t) \qquad (40)$$

$$\tau(t) = -\mu_2(t) \qquad (41)$$

Consequently, from inequality Eq. (38), it follows that:

$$\dot{V}_1(t) \leq -\beta k_3 \eta_3^2(t) a(t) - \beta k_4 \eta_4^2(t) b(t) + D. \qquad (42)$$

**Theorem 1:** Under Assumptions Eqs. (1) and (2), robust control laws in Eqs. (40) and (41) can ensure the boundedness of the closed-loop signals of the overhead crane bridge system in Eqs. (5) and (13) subject to disturbances. The closed-loop system is exponentially stable with output constraints of the middle rigid body, which means that $\lim_{t \to \infty} w(x, t) = \lim_{t \to \infty} \varphi(x, t) = 0$ holds for $\forall x \in [0, l]$, and $|w_0(t)| < k_1$ and $|\varphi_0(t)| < k_2$ hold for $\forall t \in [0, \infty)$.

**Proof:** The stability of the system can be investigated by constructing an appropriate Lyapunov function, as described below

$$V(t) = V_1(t) + V_2(t) + V_3(t) + V_4(t) \qquad (43)$$

where

$$V_2(t) = \frac{\beta}{2} \rho \int_0^{l_1} \dot{w}_L^2(x, t) \, dx + \frac{\beta}{2} K \int_0^{l_1} (\varphi_l(x, t) - w_{lx}(x, t))^2 \, dx + \frac{\beta}{2} I_P \int_0^{l_1} \dot{\varphi}_L^2(x, t) \, dx$$

$$+ \frac{\beta}{2} EI \int_0^{l_1} \varphi_{lx}^2(x, t) \, dx + \frac{\beta}{2} \rho \int_{l_1}^{l} \dot{w}_R^2(x, t) \, dx \qquad (44)$$

$$+ \frac{\beta}{2} K \int_{l_1}^{l} (\varphi_R(x, t) - w_{Rx}(x, t))^2 \, dx$$

$$+ \frac{\beta}{2} I_p \int_{l_1}^{l} \dot{\varphi}_R^2(x, t) \, dx + \frac{\beta}{2} EI \int_{l_1}^{l} \varphi_{Rx}^2(x, t) \, dx$$

$$V_3(t) = \alpha\rho \int_0^{l_1} \dot{w}_l(x,t) w_l(x,t)\,\mathrm{d}x + \alpha I_P \int_0^{l_1} \dot{\varphi}_l(x,t)\varphi_l(x,t)\,\mathrm{d}x$$

$$+ \alpha\rho \int_{l_1}^{l} \dot{w}_R(x,t) w_R(x,t)\,\mathrm{d}x + \alpha I_P \int_{l_1}^{l} \dot{\varphi}_R(x,t)\varphi_R(x,t)\,\mathrm{d}x \tag{45}$$

$$V_4(t) = \frac{\alpha\gamma_1}{2} \int_0^{l_1} w_l^2(x,t)\,\mathrm{d}x + \frac{\alpha\gamma_2}{2} \int_0^{l_1} \varphi_l^2(x,t)\,\mathrm{d}x$$

$$+ \frac{\alpha\gamma_1}{2} \int_{l_1}^{l} w_R^2(x,t)\,\mathrm{d}x + \frac{\alpha\gamma_2}{2} \int_{l_1}^{l} \varphi_R^2(x,t)\,\mathrm{d}x \tag{46}$$

In order to confirm that $V(t)$ is positive definite, we introduce the following definition

$$\chi(t) = \int_0^{l_1} \dot{w}_l^2(x,t)\,\mathrm{d}x + \int_0^{l_1} \dot{\varphi}_l^2(x,t)\,\mathrm{d}x + \int_0^{l_1} \varphi_{lx}^2(x,t)\,\mathrm{d}x$$

$$+ \int_0^{l_1} w_{lx}^2(x,t)\,\mathrm{d}x + \int_0^{l_1} \varphi_l^2(x,t)\,\mathrm{d}x + \int_0^{l_1} w_l^2(x,t)\,\mathrm{d}x$$

$$+ \int_{l_1}^{l} \dot{w}_R^2(x,t)\,\mathrm{d}x + \int_{l_1}^{l} \dot{\varphi}_R^2(x,t)\,\mathrm{d}x + \int_{l_1}^{l} \varphi_{Rx}^2(x,t)\,\mathrm{d}x$$

$$+ \int_{l_1}^{l} w_{Rx}^2(x,t)\,\mathrm{d}x + \int_{l_1}^{l} \varphi_R^2(x,t)\,\mathrm{d}x + \int_{l_1}^{l} w_R^2(x,t)\,\mathrm{d}x \tag{47}$$

Using the information from Eqs. (44) and (45) in conjunction with Lemma 1, we can draw the following conclusions:

$$V_2(t) + V_4(t) \le \frac{\beta}{2}\rho \int_0^{l_1} \dot{w}_l^2(x,t)\,\mathrm{d}x + \frac{\alpha\gamma_1}{2} \int_0^{l_1} w_l^2(x,t)\,\mathrm{d}x + \beta K \int_0^{l_1} w_{lx}^2(x,t)\,\mathrm{d}x$$

$$+ \frac{\beta}{2}EI \int_0^{l_1} \varphi_{lx}^2(x,t)\,\mathrm{d}x + \left(\beta K + \frac{\alpha\gamma_2}{2}\right) \int_0^{l_1} \varphi_l^2(x,t)\,\mathrm{d}x$$

$$+ \frac{\alpha\gamma_1}{2} \int_{l_1}^{l} w_R^2(x,t)\,\mathrm{d}x + \beta K \int_{l_1}^{l} w_{Rx}^2(x,t)\,\mathrm{d}x + \frac{\beta}{2}I_P \int_{l_1}^{l} \dot{\varphi}_R^2(x,t)\,\mathrm{d}x$$

$$+ \frac{\beta}{2}EI \int_{l_1}^{l} \varphi_{Rx}^2(x,t)\,\mathrm{d}x + \left(\beta K + \frac{\alpha\gamma_2}{2}\right) \int_{l_1}^{l} \varphi_R^2(x,t)\,\mathrm{d}x$$

$$+ \frac{\beta}{2}I_P \int_0^{l_1} \dot{\varphi}_l^2(x,t)\,\mathrm{d}x + \frac{\beta}{2}\rho \int_{l_1}^{l} \dot{w}_R^2(x,t)\,\mathrm{d}x$$

$$\le \lambda_2 \chi(t) \tag{48}$$

where $\lambda_2 = \max\left\{\frac{1}{2}\beta\rho, \frac{\beta}{2}I_P, \frac{\beta}{2}EI, \beta K + \frac{\alpha\gamma_2}{2}, \frac{\alpha\gamma_1}{2}\right\} > 0$. By applying Eq. (45), we derive the subsequent inequality:

$$|V_3(t)| \le \frac{\alpha\rho}{2} \int_0^{l_1} \dot{w}_l^2(x,t)\,\mathrm{d}x + \frac{\alpha\rho}{2} \int_0^{l_1} w_l^2(x,t)\,\mathrm{d}x + \frac{\alpha I_P}{2} \int_0^{l_1} \dot{\varphi}_l^2(x,t)\,\mathrm{d}x$$

$$+ \frac{\alpha I_P}{2} \int_0^{l_1} \varphi_l^2(x,t)\,\mathrm{d}x$$

$$+ \frac{\alpha\rho}{2} \int_{l_1}^{l} \dot{w}_R^2(x,t)\,\mathrm{d}x + \frac{\alpha\rho}{2} \int_{l_1}^{l} w_R^2(x,t)\,\mathrm{d}x + \frac{\alpha I_P}{2} \int_{l_1}^{l} \dot{\varphi}_R^2(x,t)\,\mathrm{d}x$$

$$+ \frac{\alpha I_P}{2} \int_{l_1}^{l} \varphi_R^2(x,t)\,\mathrm{d}x$$

$$\le \lambda_3 \chi(t) \tag{49}$$

in which $\lambda_3 = \max\left\{\frac{1}{2}\alpha I_P, \frac{1}{2}\alpha\rho\right\} > 0$.

By appropriately choosing the values of $\alpha$ and $\beta$ such that $\lambda_1 > \lambda_3$, the following result can be established:

$$0 < (\lambda_1 - \lambda_3)\,\chi(t) \leq V_2(t) + V_3(t) + V_4(t) \leq (\lambda_2 + \lambda_3)\,\chi(t). \tag{50}$$

Then we can further attain

$$0 < \lambda_4\,(\chi(t) + V_1(t)) \leq V(t) \leq \lambda_5\,(\chi(t) + V_1(t)) \tag{51}$$

where, $\kappa_1 = \lambda_1 - \lambda_3 > 0$ and $\kappa_2 = \lambda_2 + \lambda_3 > 0$, with $\lambda_4$ defined as the minimum of $\kappa_1, 1$, ensuring it is greater than zero. Additionally, $\lambda_5$ is defined as the maximum of $\kappa_2, 1$, also ensuring it is greater than zero.

By taking the derivative of Eq. (44) with respect to time and then substituting in Eqs. (5) through (8), we arrive at the following formulation:

$$\begin{aligned}
\dot{V}_2(t) =& \beta K \dot{w}_L(l_1, t)\,(w_{lx}(l_1, t) - \varphi_L(l_1, t)) + \beta EI \dot{\varphi}_L(l_1, t)\,\varphi_{lx}(l_1, t) \\
& - \beta K \dot{w}_R(l_1, t)\,(w_{Rx}(l_1, t) - \varphi_R(l_1, t)) - \beta EI \dot{\varphi}_R(l_1, t)\,\varphi_{Rx}(l_1, t) \\
& - \beta\gamma_1 \int_0^{l_1} \dot{w}_L^2(x, t)\,\mathrm{d}x - \beta\gamma_2 \int_0^{l_1} \dot{\varphi}_l^2(x, t)\,\mathrm{d}x - \beta\gamma_1 \int_{l_1}^{l} \dot{w}_R^2(x, t)\,\mathrm{d}x \\
& - \beta\gamma_2 \int_{l_1}^{L} \dot{\varphi}_R^2(x, t)\,\mathrm{d}x
\end{aligned} \tag{52}$$

When we substitute Eqs. (26) through (29) into the derivative of Eq. (45) and apply integration by parts, we obtain the following result:

$$\begin{aligned}
\dot{V}_3(t) =& \alpha K w_l(l_1, t)\,w_{lx}(l_1, t) - \alpha K \int_0^{l_1} w_{lx}^2(x, t)\,\mathrm{d}x - \alpha K w_l(l_1, t)\,\varphi_l(l_1, t) \\
& + \alpha\rho \int_0^{l_1} \dot{w}_l^2(x, t)\,\mathrm{d}x - \alpha\gamma_1 \int_0^{l_1} w_l(x, t)\dot{w}_l(x, t)\,\mathrm{d}x + \alpha EI \varphi_l(l_1, t)\,\varphi_{lx}(l_1, t) \\
& - \alpha EI \int_0^{l_1} \varphi_{lx}^2(x, t)\,\mathrm{d}x + 2\alpha K \int_0^{l_1} \varphi_l(x, t) w_{lx}(x, t)\,\mathrm{d}x - \alpha K \int_0^{l_1} \varphi_l^2(x, t)\,\mathrm{d}x \\
& - \alpha\gamma_2 \int_0^{l_1} \varphi_l(x, t)\dot{\varphi}_l(x, t)\,\mathrm{d}x - \alpha K w_r(l_1, t)\,w_{rx}(l_1, t) - \alpha K \int_{l_1}^{l} w_{rx}^2(x, t)\,\mathrm{d}x \\
& + \alpha K w_r(l_1, t)\,\varphi_r(l_1, t) + \alpha\rho \int_{l_1}^{l} \dot{w}_r^2(x, t)\mathrm{d}x - \alpha\gamma_1 \int_{l_1}^{l} w_r(x, t)\dot{w}_r(x, t)\,\mathrm{d}x \\
& - \alpha EI \varphi_r(l_1, t)\,\varphi_{rx}(l_1, t) - \alpha EI \int_{l_1}^{l} \varphi_{rx}^2(x, t)\,\mathrm{d}x + 2\alpha K \int_{l_1}^{l} \varphi_r(x, t) w_{rx}(x, t)\,\mathrm{d}x \\
& - \alpha K \int_{l_1}^{l} \varphi_r^2(x, t)\,\mathrm{d}x + \alpha I_p \int_{l_1}^{l} \dot{\varphi}_r^2(x, t)\mathrm{d}x - \alpha\gamma_2 \int_{l_1}^{l} \varphi_r(x, t)\dot{\varphi}_r(x, t)\,\mathrm{d}x \\
& + \alpha I_p \int_0^{l_1} \dot{\varphi}_l^2(x, t)\,\mathrm{d}x
\end{aligned} \tag{53}$$

The time derivative of $\dot{V}_4(t)$ is expected to yield the following expression:

$$\begin{aligned}
\dot{V}_4(t) =& \alpha\gamma_1 \int_0^{l_1} w_l(x, t)\dot{w}_l(x, t)\,\mathrm{d}x + \alpha\gamma_2 \int_0^{l_1} \varphi_l(x, t)\dot{\varphi}_l(x, t)\,\mathrm{d}x \\
& + \alpha\gamma_1 \int_{l_1}^{L} w_R(x, t)\dot{w}_R(x, t)\,\mathrm{d}x + \alpha\gamma_2 \int_{l_1}^{L} \varphi_R(x, t)\dot{\varphi}_R(x, t)\,\mathrm{d}x
\end{aligned} \tag{54}$$

Combining inequality Eq. (42), Eqs. (52)–(54) and utilizing Lemma 2 yield:

$$
\begin{aligned}
\dot{V}(t) \leq & -(\beta\gamma_1 - \alpha\rho)\int_0^{l_1}\dot{w}_l^2(x,t)\,\mathrm{d}x - \alpha K\int_0^{l_1}w_{lx}^2(x,t)\,\mathrm{d}x \\
& -(\beta\gamma_2 - \alpha I_P)\int_0^{l_1}\dot{\varphi}_l^2(x,t)\,\mathrm{d}x - \alpha EI\int_0^{l_1}\varphi_{lx}^2(x,t)\,\mathrm{d}x - \alpha K\int_0^{l_1}\varphi_l^2(x,t)\,\mathrm{d}x \\
& + 2\alpha K\int_0^{l_1}\varphi_l(x,t)w_{lx}(x,t)\,\mathrm{d}x - (\beta\gamma_1 - \alpha\rho)\int_{l_1}^{l}\dot{w}_R^2(x,t)\,\mathrm{d}x \\
& -(\beta\gamma_2 - \alpha I_P)\int_{l_1}^{l}\dot{\varphi}_R^2(x,t)\,\mathrm{d}x - \alpha EI\int_{l_1}^{l}\varphi_{Rx}^2(x,t)\,\mathrm{d}x - \alpha K\int_{l_1}^{l}\varphi_R^2(x,t)\,\mathrm{d}x \\
& + 2\alpha K\int_{l_1}^{l}\varphi_R(x,t)w_{Rx}(x,t)\,\mathrm{d}x - \beta k_3\eta_3^2(t)a(t) - \beta k_4\eta_4^2(t)b(t) \\
& - \alpha k_1 w_0^2(t) - \alpha k_2\varphi_0^2(t) - \alpha K\int_{l_1}^{l}w_{Rx}^2(x,t)\,\mathrm{d}x
\end{aligned}
\tag{55}
$$

From Eq. (55), it can be further obtained

$$
\begin{aligned}
V \leq & -(\beta\gamma_1 - \alpha\rho)\int_0^{l_1}\dot{w}_l^2(x,t)\,\mathrm{d}x - (\alpha K - 2\alpha K\delta_1)\int_0^{l_1}w_{lx}^2(x,t)\,\mathrm{d}x \\
& -(\beta\gamma_1 - \alpha\rho)\int_0^{l_1}\dot{w}_l^2(x,t)\,\mathrm{d}x - (\alpha K - 2\alpha K\delta_1)\int_0^{l_1}w_{lx}^2(x,t)\,\mathrm{d}x \\
& -(\beta\gamma_2 - \alpha I_P)\int_0^{l_1}-\dot{\varphi}_L^2(x,t)\,\mathrm{d}x - \left(\alpha K - \frac{2\alpha K}{\delta_1}\right)\int_0^{l_1}\varphi_l^2(x,t)\,\mathrm{d}x \\
& -(\beta\gamma_2 - \alpha I_P)\int_0^{l_1}\dot{\varphi}_l^2(x,t)\,\mathrm{d}x - \left(\alpha K - \frac{2\alpha K}{\delta_1}\right)\int_0^{l_1}\varphi_l^2(x,t)\,\mathrm{d}x \\
& - \alpha EI\int_0^{l_1}\varphi_{lx}^2(x,t)\,\mathrm{d}x - \alpha EI\int_0^{l_1}\varphi_{lx}^2(x,t)\,\mathrm{d}x
\end{aligned}
\tag{56}
$$

where, $\delta_1 > 0$ and $\delta_2 > 0$.

Utilizing Lemma 1, we have

$$
\begin{aligned}
\dot{V}(t) \leq & -(\beta\gamma_1 - \alpha\rho)\int_0^{l_1}\dot{w}_l^2(x,t)\,\mathrm{d}x - \zeta_1\int_0^{l_1}w_l^2(x,t)\,\mathrm{d}x \\
& -\left(\alpha K - 2\alpha K\delta_1 - 4l_1^2\zeta_1\right)\int_0^{l_1}w_{lx}^2(x,t)\,\mathrm{d}x \\
& -(\beta\gamma_2 - \alpha I_P)\int_0^{l_1}\dot{\varphi}_l^2(x,t)\,\mathrm{d}x - \left(\alpha K + \frac{\alpha}{8l_1^2}EI - \frac{2\alpha K}{\delta_1}\right)\int_0^{l_1}\varphi_l^2(x,t)\,\mathrm{d}x \\
& -\frac{\alpha}{2}EI\int_0^{l_1}\varphi_{lx}^2(x,t)\,\mathrm{d}x - (\beta\gamma_1 - \alpha\rho)\int_{l_1}^{l}\dot{w}_R^2(x,t)\,\mathrm{d}x - \zeta_2\int_{l_1}^{l}w_R^2(x,t)\,\mathrm{d}x \\
& -\left(\alpha K - 2\alpha K\delta_2 - 4l_2^2\zeta_2\right)\int_{l_1}^{l}w_{Rx}^2(x,t)\,\mathrm{d}x - (\beta\gamma_2 - \alpha I_P)\int_{l_1}^{l}\dot{\varphi}_R^2(x,t)\,\mathrm{d}x \\
& -\left(\alpha K + \frac{\alpha}{8l_2^2}EI - \frac{2\alpha K}{\delta_2}\right)\int_{l_1}^{l}\varphi_R^2(x,t)\,\mathrm{d}x - \frac{\alpha}{2}EI\int_{l_1}^{l}\varphi_{Rx}^2(x,t)\,\mathrm{d}x \\
& - \beta k_3\eta_3^2(t)a(t) - \beta k_4\eta_4^2(t)b(t) - \alpha k_1 w_0^2(t) - \alpha k_2\varphi_0^2(t)
\end{aligned}
\tag{57}
$$

where $\zeta_1 > 0, \zeta_2 > 0$, and $\delta_1, \delta_2, \zeta_1, \zeta_2, \alpha, \beta$ are selected to satisfy $\beta\gamma_1 - \alpha\rho > 0$, $\beta\gamma_2 - \alpha I_P > 0$, $\alpha K + \frac{\alpha}{8L_1^2}EI - \frac{2\alpha K}{\delta_1} > 0$, $\alpha K + \frac{\alpha}{8L_2^2}EI - \frac{2\alpha K}{\delta_2} > 0$, $\alpha K - 2\alpha K\delta_1 - 4L_1^2\zeta_1 > 0$, $\alpha K - 2\alpha K\delta_2 - 4L_2^2\zeta_2 > 0$.

Then $\dot{V}(t)$ can be rewritten as

$$\dot{V}(t) \leq -\lambda_6 \left( \chi(t) + V_1(t) \right) \tag{58}$$

where

$$\lambda_6 = \min \left\{ \beta\gamma_1 - \alpha\rho, \zeta_1, \alpha K - 2\alpha K\delta_1 - 4L_1^2\zeta_1, \beta\gamma_2 - \alpha I_p, \alpha K + \frac{\alpha}{8L_1^2}EI - \frac{2\alpha K}{\delta_1}, \right.$$
$$\left. \frac{\alpha}{2}EI, \quad \zeta_2, \alpha K - 2\alpha K\delta_2 - 4L_2^2\zeta_2, \alpha K + \frac{\alpha}{8L_2^2}EI - \frac{2\alpha K}{\delta_2}, \frac{2\alpha}{\beta}, \frac{2k_3}{M}, \frac{2k_4}{J} \right\}. \tag{59}$$

From inequalities Eq. (51) and Eq. (58), we have

$$\dot{V}(t) \leq -\lambda V(t) \tag{60}$$

where $\lambda = \frac{\lambda_6}{\lambda_4} > 0$.

Multiplying Eq. (60) by $e^{\lambda t}$ yields

$$V(t) \leq V(0)e^{-\lambda t}. \tag{61}$$

Based on Eqs. (51) and (61), Lemmas 3 and Lemmas 4, we have

$$\frac{1}{l_1}\varphi_l^2(x,t) \leq \frac{1}{\lambda_4}V(0)e^{-\lambda t}. \tag{62}$$

$$\frac{1}{l_2}\varphi_R^2(x,t) \leq \frac{1}{\lambda_4}V(0)e^{-\lambda t}. \tag{63}$$

$$\frac{1}{l_1}w_l^2(x,t) \leq \frac{1}{\lambda_4}V(0)e^{-\lambda t}. \tag{64}$$

$$\frac{1}{l_2}w_R^2(x,t) \leq \frac{1}{\lambda_4}V(0)e^{-\lambda t}. \tag{65}$$

$$\frac{\beta}{2}k_1 w_0^2(t) \leq \frac{1}{\lambda_4}V(0)e^{-\lambda t}. \tag{66}$$

$$\frac{\beta}{2}k_2 \varphi_0^2(t) \leq \frac{1}{\lambda_4}V(0)e^{-\lambda t}. \tag{67}$$

Then it is obvious that

$$|w_l(x,t)|, |\varphi_l(x,t)| \leq \sqrt{\frac{l_1}{\lambda_4}V(0)e^{-\lambda t}} \tag{68}$$

$$|w_R(x,t)|, |\varphi_R(x,t)| \leq \sqrt{\frac{l_2}{\lambda_4}V(0)e^{-\lambda t}} \tag{69}$$

$$|w_0(t)| \leq \sqrt{\frac{2}{\beta k_1 \lambda_4}V(0)e^{-\lambda t}}, |\varphi_0(t)| \leq \sqrt{\frac{2}{\beta k_2 \lambda_5}V(0)e^{-\lambda t}}. \tag{70}$$

Based on Eqs. (3.43)-(3.45), it is deduced that bounded initial conditions for the system result in a bounded $V(0)$. This condition implies that both $w(x,t)$ and $\varphi(x,t)$ remain bounded for every $(x,t)$ within the domain $[0,l] \times [0,\infty)$

Furthermore, according to Eqs. (68)–(70), it is evident that

$$\lim_{t\to\infty} |w(x,t)|, \lim_{t\to\infty} |\varphi(x,t)| = 0 \text{ for } \forall x \in [0,l]. \tag{71}$$

## 4 Simulation verification

For our simulation, we utilized MATLAB/Simulink version 2022b. This version, along with any later releases, is fully compatible with our simulation code. However, earlier versions (prior to 2022b) may not work as expected. The simulations were performed on a computer with the following specifications: the operating system is Windows 10 Professional, the processor is an Intel(R) Core(TM) i7-14700KF running at 3.40 GHz, and the RAM is 32.0 GB. This setup ensures the simulations run efficiently and remain stable, providing consistent performance and reliable results across various scenarios. The specified hardware delivers the necessary computational power to handle the complex calculations and data processing involved in the simulations, ensuring smooth operation without disruptions.

To verify the effectiveness of the control strategy proposed in this paper, simulations were conducted comparing it with both a no-control strategy and a PD control-based approach. Additionally, the specific control algorithm and simulation parameters for the PD control strategy can be found in detail in reference [42]. The system's dynamics are outlined in Eqs. (5)–(13). System parameters were set as follows: $\rho = 2$ kg m$^{-1}$, $K = 0.5$ N, $\gamma_1 = 2.5$ kg/(ms), $I_p = 2$ kg m$^2$, $EI = 50$ Nm, $\gamma_2 = 2.5$ kg/(ms), $M = 60$ kg, $J = 2$ kg m$^2$, $L_1 = 2$ m, $L_2 = 2$ m, and $L = 4$ m. Initial conditions were chosen for $w(x,0)$ as a piecewise function defined by the intervals of $L_1$ and $L_2$, and $\varphi(x,0) = 0$, with both $\dot{w}(x,0) = 0$ and $\dot{\varphi}(x,0) = 0$. External disturbances were modeled as $d_1(t) = 0.05 \sin(0.1t)$ and $d_2(t) = 5 \times 10^{-4} \sin(0.5t)$.

The control strategies outlined in Eqs. (40)–(41) were applied, featuring control gains $k_1 = 0.1$, $k_2 = 2 \times 10^{-3}$, $k_3 = 0.1$, $k_4 = 0.1$, $k_5 = 100$, $k_6 = 100$, and the parameters $\alpha = 1$ and $\beta = 10$. Additionally, the upper limits for the control outputs $w_0(t)$ and $\varphi_0(t)$ are set by $k_1$ and $k_2$, respectively. The simulation aims to confirm that the control design adequately restricts system responses within desired limits, even in the presence of specified disturbances.

Figures 1–3 show the displacement response of the FMS system under different control strategies. These images depict the displacement as it changes over time (horizontal axis, in seconds) and space (vertical axis, in meters). Figure 1 demonstrates the system's dynamic displacement over time without any control laws. From this figure, it is evident that the displacement fluctuation becomes more pronounced in the later stages, with significant instability, especially in the areas of intense displacement variation. Figure 2 shows the dynamic displacement response of the system under the PD boundary control method. Compared to Figure 1, the displacement fluctuations in Figure 2 are clearly reduced, indicating that the PD control method has a positive effect on reducing instability. Figure 3 presents the system's dynamic displacement response under the proposed control laws. As shown in the figure, the system's response becomes smoother, with further reduction in displacement fluctuations, and the system's stability is significantly improved. By examining Figures 1 to 3, it is evident that control laws play a crucial role in the management of dynamic systems. Our proposed control strategy markedly improves the dynamic behavior of the system, minimizes instabilities, and enhances the system's overall operational performance. Particularly, Figure 3 vividly demonstrates how the proposed control law successfully mitigates displacement fluctuations and

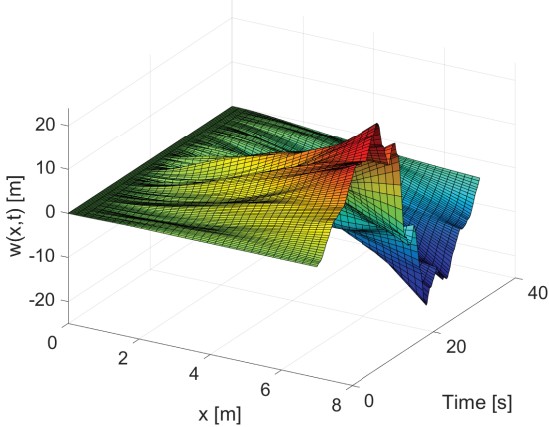

**Fig 1. The displacement $w(x,t)$ of the FMSs under the no control laws.**

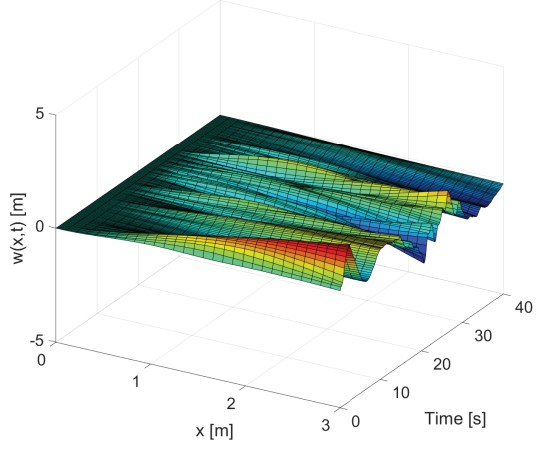

**Fig 2. The displacement $w(x,t)$ of the FMSs under the PD control laws.**

achieves sustained stability. This underscores the effectiveness of our approach in maintaining robust system performance under varying conditions.

Figure 4 displays the end displacement response of the FMS system under no control, PD control strategy, and the control strategy proposed in this paper. The black solid line represents the end displacement response without any control, showing that the system experiences significant fluctuations throughout the simulation time. The blue dashed line indicates the end displacement response under the PD control strategy, where fluctuations are notably reduced compared to the uncontrolled state. The red dashed line shows the displacement response under the proposed control strategy, where fluctuations are significantly reduced, and the overall trend tends towards a more stable level (close to 0 meters). Figure 5 presents the control inputs under different control strategies (no control, PD control, and the control strategy proposed in this paper). The black solid line shows the control input curve without any control strategy, the blue dashed line shows the control input curve under the PD control

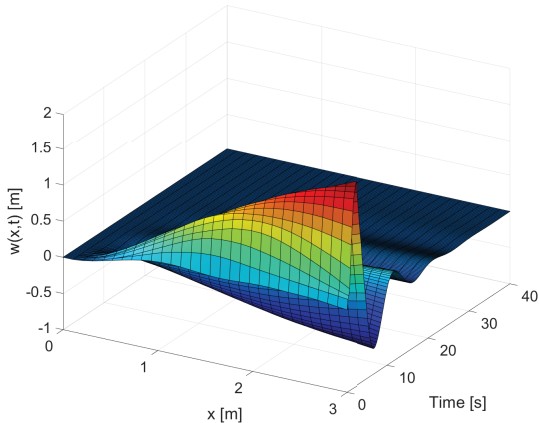

**Fig 3. The displacement $w(x, t)$ of the FMSs under the proposed control laws.**

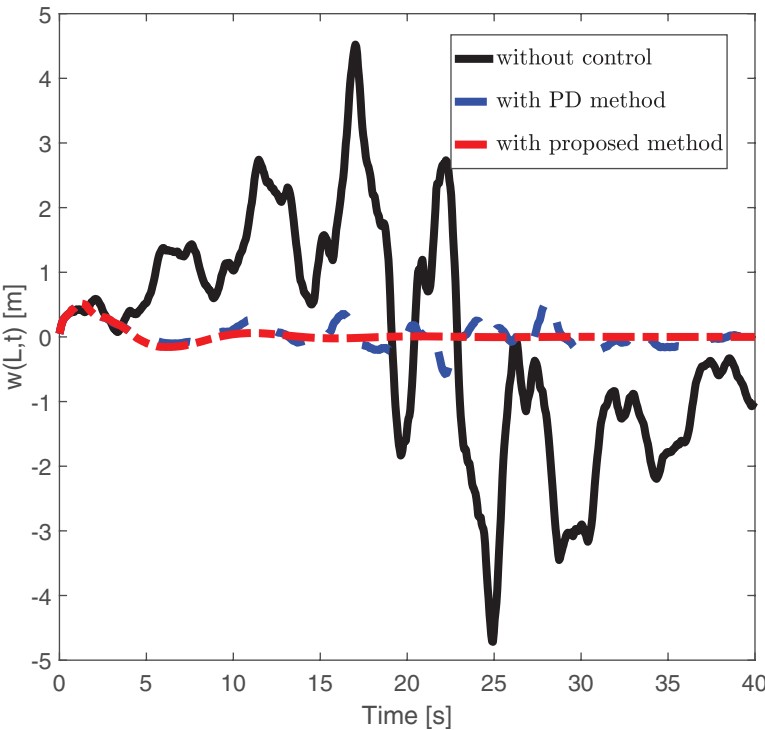

**Fig 4. The displacement $w(l, t)$ of the FMSs under the three control laws (no control method, PD control method and proposed control method) .**

strategy, and the red dashed line represents the control inputs under the proposed strategy. From Figure 5, it is evident that the control inputs remain very close to zero throughout the time period, with almost no fluctuations.

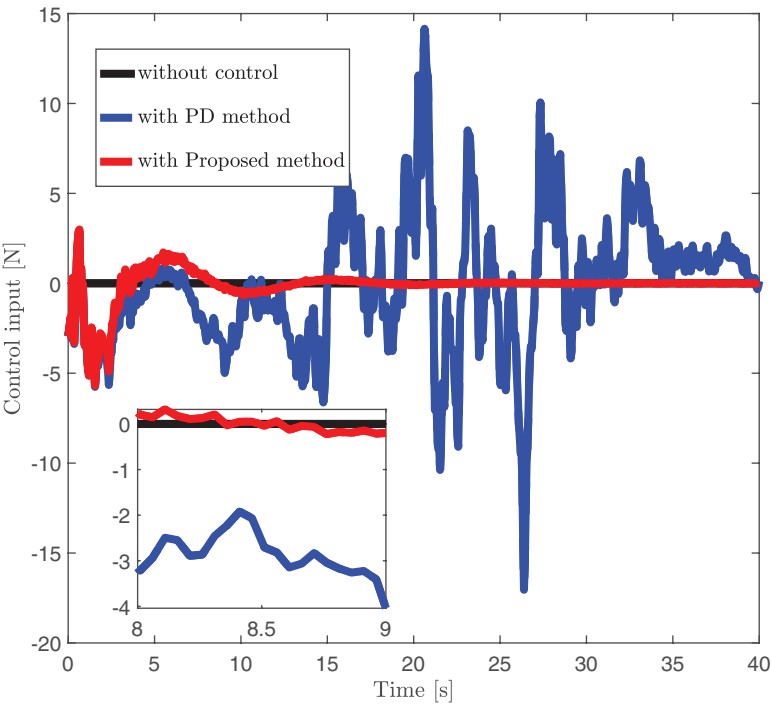

**Fig 5. Control input display diagram under three control methods (no control method, PD control method and proposed control method.**

Figures 4 and 5 convincingly showcase the benefits of our control strategies. In the absence of a control strategy, the system experiences significant instability and displacement variations. Introducing the PD control strategy considerably dampens these fluctuations. Our new control strategy goes further, stabilizing the displacement even more and steering it towards an equilibrium condition. Furthermore, the control input diagram shown in Figure 5 reveals that our proposed method stabilizes the system with minimal additional control force, leading to efficient and stable operation. This highlights the dual benefits of our approach, achieving system stability with reduced energy requirements, thereby enhancing both effectiveness and cost-efficiency.

## 5 Discussion and future work

This paper introduces a vibration suppression technology for FMSs, employing a robust control scheme formulated from a PDEs model and Barrier Lyapunov Functions (BLF). The scheme effectively reduces vibrations and rotations, with a significant decrease in displacement (up to 75%) and rotational oscillations (up to 60%), enhancing the precision and stability of equipment, and extending its operational lifespan. This technology finds application across diverse sectors such as aerospace, robotics, and precision manufacturing, where precise control is crucial. The control laws, executed through backstepping technology, maintain system outputs within acceptable ranges, even amid actuator faults and environmental disturbances, ensuring robust system stability. Comprehensive simulations have validated the effectiveness of this control approach, demonstrating its capability to mitigate displacement and rotation under uncertain disturbances while complying with output constraints.

Although the current control scheme shows excellent performance, the potential for generalization and scalability to other types of FMS or broader industrial contexts presents a compelling avenue for future research. We plan to explore the adaptability of the proposed control strategies to different manufacturing settings, considering various machine architectures and operational conditions.

## Supporting information

S1. Paper program.
(PDF)

## Author contributions

**Data curation:** Yuzhi Tang.

**Methodology:** Yuzhi Tang.

**Writing – original draft:** Yuzhi Tang.

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
