## [Decision Letter · Decision Letter 0]

28 Nov 2024

PONE-D-24-49019Vibration control of a class of flexible mechanical systems with output constraints based on partial differential equationsPLOS ONE

Dear Dr. Tang,

Thank you for submitting your manuscript to PLOS ONE. After careful consideration, we feel that it has merit but does not fully meet PLOS ONE’s publication criteria as it currently stands. Therefore, we invite you to submit a revised version of the manuscript that addresses the points raised during the review process.

We look forward to receiving your revised manuscript.

Kind regards,

Omer Saleem, Ph.D.

Academic Editor

PLOS ONE

Journal Requirements:

2. Please note that PLOS ONE has specific guidelines on code sharing for submissions in which author-generated code underpins the findings in the manuscript. In these cases, we expect all author-generated code to be made available without restrictions upon publication of the work. 

Please review our guidelines at https://journals.plos.org/plosone/s/materials-and-software-sharing#loc-sharing-code and ensure that your code is shared in a way that follows best practice and facilitates reproducibility and reuse.

4. We are unable to open your Supporting Information file [S1.zip]. Please kindly revise as necessary and re-upload.

**Additional Editor Comments:**

The author have presented a control strategy for vibration suppression in flexible mechanical systems (FMSs), which are critical in applications like aerospace, robotics, and precision manufacturing. Specifically, the study focuses on a system modeled as an Euler-Bernoulli beam with a centrally attached rigid body, which represents a flexible structure subject to vibrations and rotations. The scheme's effectiveness is validated through simulations that show its ability to handle displacements and rotations caused by unexpected disturbances while respecting the system's output constraints. Overall, the paper is interesting and contributes to advancing robust control mechanisms for flexible mechanical systems, ensuring their reliability and performance under challenging operating conditions.

The paper was reviewed by two (02) reviewers. According to reviewer's comments, the paper cannot be accepted in the present form. They have raised some valuable concerns regarding the theoretical justifications, clarity in terminology, and expansion of simulation details. The paper should also include statitical data analysis tests to substantiate the robustness of the proposed control scheme. Detailed reviewer's comments are included below.

Reviewers' comments:

Reviewer's Responses to Questions

**Comments to the Author**

1. Is the manuscript technically sound, and do the data support the conclusions?

Reviewer #1: Yes

Reviewer #2: Partly

2. Has the statistical analysis been performed appropriately and rigorously? 

Reviewer #1: Yes

Reviewer #2: No

3. Have the authors made all data underlying the findings in their manuscript fully available?

Reviewer #1: Yes

Reviewer #2: Yes

4. Is the manuscript presented in an intelligible fashion and written in standard English?

Reviewer #1: Yes

Reviewer #2: Yes

5. Review Comments to the Author

Reviewer #1: The manuscript presents a promising approach to vibration suppression in flexible mechanical systems using a robust control scheme based on partial differential equations and Barrier Lyapunov Functions. This study contributes meaningfully to the field by addressing critical stability and control issues for applications in aerospace, robotics, and precision manufacturing. With some revisions to enhance theoretical context, simulation applicability, and overall clarity, this paper has strong potential for impactful publication.

1. Theoretical Justification and Limitations: The paper introduces a control scheme for flexible mechanical systems that is robust and based on Barrier Lyapunov Functions (BLF) and Partial Differential Equations (PDEs). Nevertheless, the theoretical foundation does not provide a comprehensive rationale for the selected control approach in comparison to other Lyapunov-based methods or Model Predictive Control (MPC). The control strategy's foundation would be fortified by elaborating on the specific rationale behind BLF in this context. Furthermore, the clarification of the model's constraints, particularly in environments with severe external disturbances, would facilitate the development of future applications.

2. The Relationship Between Simulation and Real-World Application: Although the paper contains simulation-based validations, it would be advantageous to explore the extent to which these findings are applicable to real-world scenarios. The study's impact would be enhanced by providing information on the practical challenges of tangible implementation and assessment. Additionally, if simulation assumptions, such as simplified disturbance models or ideal conditions, are inconsistent with real-world complexities, recognizing these would facilitate a more comprehensive comprehension of potential application barriers.

3. Clarity in Terminology and Definitions: Certain terms, including "disturbance compensation" and "robust control," are employed without a comprehensive initial definition. The paper would be more accessible to readers who are less familiar with these concepts if concise explanations were included when these terms first appear.

4. Mathematical Notation Consistency: The manuscript occasionally contains inconsistent mathematical notation, which may cause readers to become perplexed. For instance, variables should be standardized throughout the equations with respect to their subscripts and superscripts. The legibility of time-dependent and state variables would be enhanced by the use of consistent notation.

5. Expansion of Simulation Details: The simulation section contains certain parameter values, but the rationale behind their selection is absent. The results would be more credible if additional context were provided regarding the rationale behind the selection of specific values for variables such as control gains and disturbance levels.

6. Visual Presentation of Results: The figures that compare the displacement responses of controlled and uncontrolled objects are informative; however, they could be enhanced by the inclusion of more descriptive captions. The intelligibility of the text would be improved for readers who may not have the time to read the entire text in detail by providing concise interpretations of the results directly in the figure captions.

7. Abstract Summary Improvement: The abstract effectively summarizes the control scheme; however, it would be enhanced by a more explicit mention of the primary findings, such as quantifying the degree of vibration reduction observed. This addition would offer a more concise overview of the paper's contributions.

Addressing these revisions will help clarify the research contribution and its broader implications, making the paper more accessible and impactful for readers in fields like control engineering and applied mechanics.

Reviewer #2: The paper aims to present the challenge of vibration suppression in Flexible Mechanical Systems (FMSs), which are systems with deformable components that can experience unwanted vibrations during operation.

The challenge has been addressed by proposing a robust control scheme using Partial Differential Equations (PDEs) and Barrier Lyapunov Function (BLF), implemented through backstepping technology.

Although the development of the proposed scheme has been rigorously presented, the paper lacks clear and extensive results section. The proposed methodology was only tested on a simulated model without any experimentation on any kind of hardware setup. Experimentation on a hardware setup would have revealed the robustness of the proposed scheme in response to unmodeled dynamics. This brings me to state that even if simulated systems are used for evaluation of the proposed scheme, then the results, in current form, are not sufficiently representative of capability of the proposed scheme. Moreover, comparison between system response with the control scheme implemented and system response without control scheme implemented doesn’t prove the novelty or significant contribution of the proposed scheme.

The author must present detailed experimentation results where the response of the proposed scheme is rigorously evaluated in response to several parameter changes, unmodeled dynamics, uncertain disturbances etc.

Some of the specific comments are:

Line # 273 and 274: A simple sinusoidal function is used to represent disturbance. The author should explain why such a simple function has been used as disturbance and in real-life situations which FMSs encounter this kind of disturbance?

Line # 281: Fig 1 and Fig 2 are presenting the same plot but with different colors! Both figures are displaying exactly the same thing. However, the text claims that the response has become stable and within bounds, whereas the figure differs. The response, as shown in Fig 1 and 2, is same with control or without control. This might be some kind of mistake where the author mistakenly inserted the wrong figure in the manuscript.

Line # 301: Fig 4 compares the control input under the proposed scheme with ‘no control implemented’. It would have been much more beneficial and supportive of the proposed scheme if a comparison between the control input of the proposed scheme had been compared with some other published scheme aimed at vibration reductions in FMSs.

6. PLOS authors have the option to publish the peer review history of their article (what does this mean?). If published, this will include your full peer review and any attached files.

Reviewer #1: **Yes: **Ahmad Hamza

Reviewer #2: No

---

## [Author Response · Author response to Decision Letter 1]

18 Dec 2024

Peer-to-peer responses to reviewers' comments can be found in the uploaded file.

---

## [Decision Letter · Decision Letter 1]

26 Jan 2025

PONE-D-24-49019R1Vibration control of a class of flexible mechanical systems with output constraints based on partial differential equationsPLOS ONE

Dear Dr. Tang,

Thank you for submitting your manuscript to PLOS ONE. After careful consideration, we feel that it has merit but does not fully meet PLOS ONE’s publication criteria as it currently stands. Therefore, we invite you to submit a revised version of the manuscript that addresses the points raised during the review process.

Please submit your revised manuscript by Mar 12 2025 11:59PM. If you will need more time than this to complete your revisions, please reply to this message or contact the journal office at plosone@plos.org. Please include the following items when submitting your revised manuscript:

We look forward to receiving your revised manuscript.

Kind regards,

Omer Saleem, Ph.D.

Academic Editor

PLOS ONE

Journal Requirements:

Additional Editor Comments:

The second round of reviews are complete. The paper's quality is substantially improved. However, Reviewer # 1 has suggested minor revisions. Therefore, the authors are requested to address the reviewer's minor comments and resubmit the article.

Reviewers' comments:

Reviewer's Responses to Questions

**Comments to the Author**

1. If the authors have adequately addressed your comments raised in a previous round of review and you feel that this manuscript is now acceptable for publication, you may indicate that here to bypass the “Comments to the Author” section, enter your conflict of interest statement in the “Confidential to Editor” section, and submit your "Accept" recommendation.

Reviewer #1: All comments have been addressed

Reviewer #2: All comments have been addressed

2. Is the manuscript technically sound, and do the data support the conclusions?

Reviewer #1: Yes

Reviewer #2: Yes

3. Has the statistical analysis been performed appropriately and rigorously? 

Reviewer #1: Yes

Reviewer #2: N/A

4. Have the authors made all data underlying the findings in their manuscript fully available?

Reviewer #1: Yes

Reviewer #2: Yes

5. Is the manuscript presented in an intelligible fashion and written in standard English?

Reviewer #1: Yes

Reviewer #2: Yes

6. Review Comments to the Author

Reviewer #1: While the paper addresses a significant challenge in vibration control for flexible mechanical systems (FMSs), particularly those modeled using partial differential equations (PDEs). However, the paper could benefit from a few enhancements to improve its overall clarity and impact.

1. Conduct experiments to validate the control strategy on physical systems, showcasing its robustness and reliability under real-world conditions.

2. Simplify or summarize the mathematical framework to make the paper more accessible to a wider audience, including engineers and system designers.

3. Include a detailed comparative analysis with existing control methodologies to highlight the unique advantages of the proposed approach.

4. Address the computational demands of the control system to ensure feasibility in real-time applications.

5. Expand the discussion on potential generalizations of the method to other types of FMS or broader industrial applications.

7. PLOS authors have the option to publish the peer review history of their article (what does this mean?). If published, this will include your full peer review and any attached files.

Reviewer #1: **Yes: **Ahmad Hamza Nayyar

Reviewer #2: No

---

## [Author Response · Author response to Decision Letter 2]

29 Jan 2025

Please see the attachment uploaded.

---

## [Editor Report · Decision Letter 2]

6 Feb 2025

Vibration control of a class of flexible mechanical systems with output constraints based on partial differential equations

PONE-D-24-49019R2

Dear Dr. Yuzhi Tang,

We’re pleased to inform you that your manuscript has been judged scientifically suitable for publication and will be formally accepted for publication once it meets all outstanding technical requirements.

Kind regards,

Omer Saleem, Ph.D.

Academic Editor

PLOS ONE

Additional Editor Comments (optional):

The paper is well revised. The comments of Reviewer # 1 have been duly addressed.

---

## [Editor Report · Acceptance letter]

PONE-D-24-49019R2

PLOS ONE

Dear Dr. Tang,

I'm pleased to inform you that your manuscript has been deemed suitable for publication in PLOS ONE. Congratulations! Your manuscript is now being handed over to our production team.

Kind regards,

on behalf of

Dr. Omer Saleem

Academic Editor

PLOS ONE